

# Individual and flock immunity responses of naïve ducks on smallholder farms after vaccination with H5N1 Avian Influenza vaccine: a study in a province of the Mekong Delta, Vietnam

Hoa Thi Thanh Huynh[1], Liem Tan Truong[2], Tongkorn Meeyam[3,4], Hien Thanh Le[5] and Veerasak Punyapornwithaya[3,6]

[1] Faculty of Veterinary Medicine, Chiang Mai University, Chiang Mai, Thailand
[2] Sub-Department of Animal Health (SDAH) Ben Tre Province, Ben Tre, Vietnam
[3] Veterinary Public Health Centre for Asia Pacific (VPHCAP) and Excellent Center of Veterinary Public Health, Faculty of Veterinary Medicine, Chiang Mai University, Chiang Mai, Thailand
[4] Department of Veterinary Biosciences and Veterinary Public Health, Faculty of Veterinary Medicine, Chiang Mai University, Chiang Mai, Thailand
[5] Department of Infectious Diseases and Veterinary Public Health, Faculty of Animal Science and Veterinary Medicine, Nong Lam University, Ho Chi Minh City, Vietnam
[6] Department of Food Animal Clinic, Faculty of Veterinary Medicine, Chiang Mai University, Chiang Mai, Thailand

Corresponding author
Veerasak Punyapornwithaya,
veerasak.p@cmu.ac.th

## ABSTRACT

In Vietnam, vaccination has played a crucial role in the national strategy for the prevention and control of H5 highly pathogenic avian influenza (HPAI). This study aimed to evaluate antibody responses of immunologically naïve domestic ducks to H5N1 avian influenza vaccine currently used in the national mass vaccination program of Vietnam. Blood samples of 166 ducks reared on smallholder farms were individually collected at three sampling time points, namely, right before vaccination, 21 days after primary vaccination, and 21 days after booster vaccination. Vaccine-induced antibody titers of duck sera were measured by the hemagglutination inhibition assay. Temporal differences in mean antibody titers were analyzed using the generalized least-squares method. No sampled ducks showed anti-H5 seropositivity pre-vaccination. The geometric mean titer (GMT) of the vaccinated ducks was 5.30 after primary vaccination, with 80% of the vaccinated ducks showing seropositivity. This result indicates that the immunity of duck flocks met the targets of the national poultry H5N1 HPAI mass vaccination program. GMT and seropositive rates of the ducks were 6.48 and 96.3%, respectively, after booster vaccination, which were significantly higher than those after primary vaccination. Flock-level seroprotection rate significantly increased from 68% to 84.7%, whereas variability in GMT titers decreased from 34.87% to 26.3%. This study provided important information on humoral immune responses of ducks to the currently used H5N1 vaccine under field conditions. Our findings may help guide veterinary authorities in planning effective vaccine protocols for the prevention and control of H5N1 in the target poultry population.

## INTRODUCTION

In Vietnam—a country with a high total poultry population—H5N1 highly pathogenic avian influenza (HPAI) has become endemic, which has resulted in tremendous economic losses to the poultry industry. This disease also poses a considerable threat to public health because it has caused sporadic human infections since its first reported outbreak in 2003 (*Bui et al., 2014*; *Desvaux et al., 2011*; *Tran et al., 2016*). The Vietnamese Government initiated a national poultry mass vaccination program against H5-type HPAI viruses in 2005 after the failure of other anti-HPAI measures such as massive stamping out, movement control, and disinfection (*Tran et al., 2016*). Vaccination has reduced the number of HPAI infections and outbreaks among poultry and has consequently reduced the risk of human exposure and number of human cases; these results are important steps toward the prevention and control of HPAI (*Swayne, 2012*). The Re-6 vaccine (A/duck/Guangdong/1332/2010 H5N1 clade 2.3.2) has been extensively used to immunize poultry since 2014. This vaccine contains antigens with close antigenic similarity to the H5N1 virus subclade 2.3.2.1c, which is widely circulated in southern Vietnamese provinces (*Le & Nguyen, 2014*). The vaccine has been used in Vietnam until now since it has been showing a certain protective effect against H5N1 HPAI to the poultry population.

Domestic ducks represent the second largest poultry population in Vietnam following chickens. Duck populations contribute to the maintenance and dissemination of the H5-type HPAI viruses because they are natural reservoirs of this virus (*Hulse-Post et al., 2005*; *Swayne & Kapczynski, 2008*). Therefore, in Vietnam, mass immunization of ducks is a part of the disease control strategy (*Cha et al., 2013*). The Mekong Delta (MKD) has a large population and high density of domestic ducks and is at high risk of H5N1 HPAI outbreaks because of the presence of large numbers of backyard or smallholder poultry farms (*Food and Agriculture Organizations of United Nations , FAO*; *Henning et al., 2011*). The small-scale poultry production system has several characteristics that have made it a primary site of H5N1 HPAI viral infections. The problem of inappropriate vaccination is prevalent among smallholder poultry farms. Moreover, farm owners lack sufficient knowledge of poultry diseases and, consequently, show poor compliance to vaccination guidelines (*Swayne, 2011*). Many ducks on smallholder farms in MKD have not been administered with the recommended two-dose vaccination regimen; instead, local farmers implement the single-dose vaccination regimen. Similarly, rates of compliance to the recommended two-dose vaccination regimen for meat-type ducks are low in southwestern regions (*Cuong et al., 2016*). Whether the desired protective antibody response can be induced by the single-dose vaccination regimen warrants further inquiry, and the benefit of the two-dose vaccination regimen should be confirmed. Nevertheless, published data on the effectiveness of vaccines against avian influenza (AI), particularly the Re-6 vaccine, in inducing antibody response in domestic ducks under field conditions remain limited (*Henning et al., 2011*; *McLaws et al., 2015*). This limitation, in turn, restricts the availability of information that may guide veterinary authorities in improving the national H5N1 HPAI vaccination strategy of Vietnam. *Kandeil et al. (2017)* assessed the immunogenicity and vaccine efficacy against an

Egyptian H5N1 clade 2.2.1.2 virus in different avian hosts in backyard conditions (*Kandeil et al., 2017*).

In this study, we aimed to evaluate the effectiveness of the Re-6 vaccine against H5N1 HPAI in immunologically naïve domestic ducks reared on smallholder farms. We evaluated the effectiveness of the vaccine in inducing immunological responses under field conditions by examining the levels and variation of antibody responses at individual and flock levels.

## MATERIALS & METHODS

### Sample size

Sample size was determined using power analysis and sample size software (PASS version 15, NCSS, LLC. Utah, USA; http://www.ncss.com/software/pass/) on the basis of statistical analysis with repeated measured data. A total of 166 vaccinated ducks from 11 smallholder duck farms were included. Five ducks from each farm received saline instead of the vaccine (control). In total, 20 ducks, which were individually identified using leg bands, were sampled from each farm.

### Study design

The study was conducted in two districts in a province of MKD, southern Vietnam, from July 2017 to December 2017. No H5-type HPAI outbreaks have been reported in the province since 2014. Ducks aged 18–20 days were selected from 11 farms with the support of the Sub-Department of Animal Health (SDAH). The 11 farms represented various flock sizes (approximately 100–1,300 ducks per farm), production purposes (meat or layer ducks), and duck breeds. Each selected duck was vaccinated twice. Blood samples of each duck were collected at three separate time points, namely, right before vaccination, 21 days after primary vaccination (21 dpv), and 21 days after booster vaccination (21 dpbv). The first blood sample was immediately collected before primary vaccination. Pre-vaccination sampling was performed to detect H5-specific antibodies that were potentially derived from maternal immunity or natural infection.

Vaccination was performed with the inactivated reassortant H5N1 avian influenza vaccine, Re-6 strain, which expresses the HA antigen of A/duck/Guangdong/1332/2010 H5N1 clade 2.3.2 (HA titer $\geq$ 1:256 before deactivation). This strain is the only anti-H5N1 HPAI vaccine used for mass immunization in the province in which this study was conducted. Vaccines were intramuscularly administered in the breast using automatic syringes. Each duck received 0.5 and 1 mL of the primary and booster vaccines, respectively. Except for five control ducks, the remainder of the flock was vaccinated.

Briefly, 1–2 mL of blood was drawn from each duck through the medial metatarsal vein. Serum was separated from blood by centrifugation. In addition to blood samples, pooled tracheal swab samples were collected from five vaccinated ducks from each farm at the final sampling time point. The swabs were sent to the Regional Animal Health Office VI (RAHO VI) and tested via real-time reverse transcriptase-polymerase chain reaction (RRT-PCR) to detect H5-type HPAI viruses circulating in the sampled farms during the observation period.

## Serological assay for the detection of H5-specific antibodies in duck sera

H5-specific antibodies in vaccinated duck sera were detected and quantified by the hemagglutination inhibition (HI) assay in V-bottom microtiter plates with two-fold dilutions, 0.5% specific pathogen-free chicken red blood cells (RBCs), and 4 hemagglutination units (HAU) of antigen derived from the H5N1 virus strain A\Ck\Scot\59 (RAA 7002; APHA Scientific, Surrey, UK). Tests were conducted by the SDAH of the province (license number LAS-NN 59; ISO/IEC 17025: VILAT-0043) in accordance with their routine HPAI post-vaccination serosurveillance method. A reference positive serum with a known titer and a negative control serum were included in each test plate. Before the serological assay, duck sera were heat-inactivated at 56 °C for 30 min and then treated with 10% chicken RBC suspension for removing nonspecific inhibitors to prevent the occurrence of nonspecific HA reactions in the sera of nonchicken species during the HI assay.

The HI titer of a sample is the reciprocal of the highest serum dilution that causes complete inhibition of HA activity of RBCs. HI titers were reported as $\log_2$ titers (*Ferreira et al., 2012*; *Cagle et al., 2012*) for compatibility with results obtained through the routine post-vaccination sero-monitoring program. The starting dilution for the HI assay was 1:8 ($3\log_2$). Samples with HI titer <4 were considered seronegative, whereas those with HI titer $\geq 4$ were considered seropositive. These thresholds are in compliance with Vietnam's national regulation on post-vaccination surveillance for H5N1 HPAI (MARD-DAH, 487/TY-DT, 2009), which is based on the OIE Manual (*OIE, 2009*). For calculating antibody geometric mean titers (GMTs), samples without detectable antibody levels (HI titer $<3\log_2$) were assigned an HI titer of $2\log_2$. Seropositive rates (%) were calculated with the cutoff level of $4\log_2$, and seroprotection rates (%) were calculated starting from $5\log_2$, following the criteria set in the OIE Manual (*OIE, 2015*).

## Ethics statement

This study was approved by the Institutional Ethical Review Board of Hanoi University of Public Health (IRB-HUPH, approval number 308/2017/YTCC-HD3). The IRB was registered with the U.S. Department of Health and Human Services (IORG number 0003239, FWA number FWA00009326). Permission to conduct the study was obtained from the SDAH of the province where this study was conducted. The SDAH also collaborated on this project.

## Statistical analysis

Antibody titers were transformed into $\log_2$ values, as mentioned above, prior to further analysis. Descriptive data of HI antibody titers were presented by individual farms and by the time of sample collection (''time''). GMT [GMT $\pm$ standard error (SE)], coefficient of variation (%CV, presenting variation in antibody titers, %CV = $100\times$ standard deviation/mean), % seropositive rate, and % seroprotection rate were calculated.

Temporal differences in GMT were compared using the generalized least-squares (GLS) method for repeated measurements using R statistical software version 3.3.2 (*R*

*Development Core Team, 2016*) with the *nlme* package (*Pinheiro, DebRoy & Sarkar, 2015*).
A mixed model with GLS was constructed and fitted using the restricted maximum
likelihood estimation method. GMT of antibodies was defined as the dependent variable.
"Farm" and "time" were defined as fixed effects, whereas "individual duck" was defined as
random effect. Various mixed models were constructed with different covariance structures,
including compound symmetry, general correlation matrix, and autoregressive process of
order1 (ar1) (*Pinheiro, DebRoy & Sarkar, 2015*). Values of the Akaike information criterion
(AIC) for each model were then compared to identify the best-fit model. The model with
compound symmetry covariance structure had the lowest AIC value. Thus, results of
this model were interpreted. Residuals from the fitted model were tested for normality
by plotting standardized residuals against quintiles of the standard normal as well as for
homogeneity of variance by plotting standardized residuals against fitted values. Tukey's
test was used for multiple comparisons when mean differences were significant. Seropositive
rates after primary and booster vaccinations were compared using the proportionality test.
The level of significance for statistical analysis was set at $\alpha = 0.05$.

## RESULTS

### Anti-H5 HI antibody titers

To investigate whether maternal antibodies have already decreased and active infections
have not occurred, pre-vaccination HI titer levels of the ducks were evaluated. No sampled
ducks were positive for pre-vaccination anti-H5 antibodies (HI titer < 4). The control ducks
did not exhibit detectable antibody levels throughout the observation period. Notably,
RRT-PCR revealed that the H5N1 virus was undetected in the samples. Moreover, H5-type
HPAI outbreaks did not occur in the study areas during the observation period, and the
vaccination did not result in adverse effects or illnesses among duck flocks. Thus, the
vaccine was well tolerated by the ducks.

HI assay results for antibody responses after each of the two H5N1 HPAI vaccinations
are summarized in Table 1. Nearly 17% of the vaccinated ducks ($n = 28/166$) did not
respond to primary vaccination (HI titers < 4), whereas more than 70% showed antibody
responses with HI titers between 4 and 7. Booster vaccination increased antibody titers,
and almost 73% of the vaccinated ducks ($n = 119/164$) showed HI titers between 6 and 9.
Thus, increased HI titers are the dominant humoral immune responses of ducks to each
dose of the Re-6 vaccine.

Antibody titers increased over time. GMT after booster vaccination was significantly
higher than that after primary vaccination (Table 1). The highest difference in GMTs was
$2.8\log_2$. This increasing trend was observed on all sampled farms (Fig. 1). In addition,
considerable variations were observed in antibody responses at 21 days after primary
vaccination. Minor variations in antibody responses following booster vaccination were
observed on all sampled farms (Fig. 1, Tables 1 and 2).

### Seropositive rates and seroprotection rates

Pre-vaccination, no sampled duck tested positive for H5N1 HPAI antibodies. Booster
vaccination significantly increased seropositive and seroprotection rates of the vaccinated

**Table 1** **Humoral immunity of vaccinated ducks at 21 days after primary vaccination (21 dpv) and 21 days after booster vaccination (21 dpbv).** Mean, variability of HI titers, proportion of vaccinated ducks showing seropositivity, and proportion of vaccinated ducks showing seroprotection are presented as values of GMT, %CV, seropositive rate (%), and seroprotection rate (%), respectively.

| Time | N | HI titer distribution ($\log_2$) | | | | | | | | GMTs (mean ± SE) | CVs (%) | Seropositive rates | Seroprotection rates |
|------|---|------|---|---|---|---|---|---|---|------|------|------|------|
| | | <3 | 3 | 4 | 5 | 6 | 7 | 8 | 9 | | | | |
| 21 dpv | 166 | 21 | 7 | 25 | 31 | 33 | 32 | 14 | 3 | 5.30 ± 0.14[a*] | 34.87 | 83.00[a*] | 68.07[a*] |
| 21 dpbv | 164 | 1 | 5 | 19 | 20 | 38 | 32 | 24 | 25 | 6.48 ± 0.13[b] | 26.30 | 96.30[b] | 84.76[b] |

**Notes.**

$N$, total number of ducks in each observation; GMT, geometric mean titer ($\log_2$) of total number of vaccinated ducks; %CV, coefficient of variation, indicating the level of variability of HI titers; Seropositive, HI titers $\geq 4$; Seroprotection, HI titers $\geq 5$.

[*ab]Values between rows with differing superscripts denote differences in mean or in proportion ($p < 0.05$).

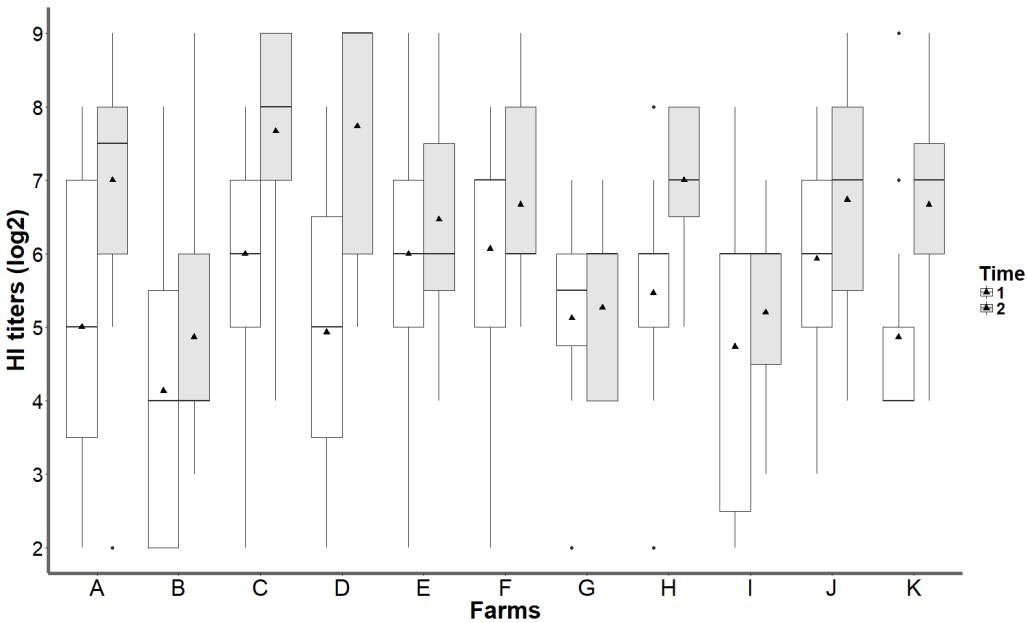

**Figure 1** **Distribution of HI titers against the H5 HPAI virus by individual farm after primary and booster vaccinations.** Farms are coded from A to K. Sampling times are referred to as "time," and "21 dpv" and "21 dpbv" represent HI results at 21 days after the primary vaccination and 21 days after the booster vaccination, respectively. GMT is represented by ▲ in the middle of each box.

ducks ($p < 0.01$) (Table 1). Seropositive rates following booster vaccination exceeded 80% on all farms. Similarly, seroprotection rates increased after booster vaccination (Table 2).

## DISCUSSION

This study was conducted to evaluate antibody responses of immunologically naïve ducks reared on smallholder farms. The vaccination protocol used in this study is similar to that currently applied by governmental veterinary services in the ongoing mass vaccination program against H5N1 HPAI.

A potential limitation of this study is that because of the evolutionary dynamics of H5-type HPAI virus clades, the strain A\Ck\Scot\59 H5 antigen might be suboptimally

**Table 2** Seropositive and seroprotection rates of ducks from the 11 sampled farms at 21 days after primary vaccination (21 dpv) and 21 days after booster vaccination (21 dpbv).

| Farm | Seropositive rate[a] 21 dpv | Seropositive rate 21 dpbv | Seroprotection rate[b] 21 dpv | Seroprotection rate 21 dpbv |
|------|------|------|------|------|
| A | 73.33 | 92.86 | 53.33 | 92.86 |
| B | 66.67 | 86.67 | 33.33 | 40.00 |
| C | 93.33 | 100.00 | 86.67 | 93.33 |
| D | 73.33 | 100.00 | 60.00 | 100.00 |
| E | 86.67 | 100.00 | 80.00 | 93.33 |
| F | 93.33 | 100.00 | 86.67 | 100.00 |
| G | 87.50 | 100.00 | 75.00 | 53.33 |
| H | 86.67 | 100.00 | 80.00 | 100.00 |
| I | 60.00 | 80.00 | 60.00 | 73.33 |
| J | 93.33 | 100.00 | 93.33 | 93.33 |
| K | 100.00 | 100.00 | 40.00 | 93.33 |

**Notes.**
[a] Seropositive rate = the proportion of seropositive vaccinated ducks (HI titers $\geq 4$).
[b] Seroprotection rate = the proportion of vaccinated ducks with HI titers at levels that protect against mortality from HPAI infection in accordance with the OIE's recommendation (HI titers $\geq 5$).

antigenically matched to the Re-6 vaccine strain. However, because of nonscientific considerations of the DAH, the strain A\Ck\Scot\59 H5 has been the only one licensed and widely used for anti-H5 antibody detection in routine HPAI post-vaccination serosurveillances and commercial tests in veterinary diagnostic laboratories throughout the country until now. The main objective of post-vaccination sero-monitoring programs is to estimate the proportion of poultry with anti-H5 antibodies. Therefore, due to the fact that several other vaccines are used in different regions of the country, the Scot/59 antigen strain has been considered reasonably effective in evaluating antibody titers induced by vaccinations. Therefore, for this study to have a practical significance that its results can be comparable with those of other relevant studies conducted in Vietnam and those obtained through routine post-vaccination sero-monitoring programs, we used the Scot/59 antigen in our study.

Given the variations in characteristics of the household farming sector, ducks may exhibit different responses to the vaccination protocol applied by the governmental veterinary services. Therefore, this study included 11 smallholder farms to represent variations in farm characteristics. Meat and layer ducks were included. Most included ducks were mixed breeds. Considering large variations in flock sizes, management practices, and other factors related to sampled farms, we could not stratify farms based on flock size or management type. Therefore, individual farms were included in the GLS model as a fixed effect.

HA-specific antibody titers measured by the HI assay were the principal indicators of vaccine-induced protective immunity against H5N1 HPAI viruses (*Suarez & Schultz-Cherry, 2000*; *Sitaras et al., 2016*). Mean antibody titers within the poultry population are expected to increase following vaccination. In this study, the two-dose vaccination regimen stimulated antibody response in ducks. Pre-vaccination, the ducks lacked HA-specific antibodies. However, numerous ducks showed immune responses after primary

vaccination. GMT values significantly increased after booster vaccination compared with those after primary vaccination. After primary vaccination, the desired overall GMT was achieved despite the lack of seroconversion in some vaccinated ducks. GMT reported in this study was higher than that reported in previous studies conducted in other provinces of MKD, involving poultry vaccinated with the same vaccine. One study has reported a GMT value of 1.63 in 28-day-old ducks (*Phan & Tran, 2016*), whereas another has reported a GMT value of 3.32 in 35-day-old ducks after primary vaccination (*Pham, 2015*). In a study conducted in the Tien Giang province, GMT values of 1.7, 3.4, 4.3, and 4.45 have been reported in 15-, 45-, 75-, and 105-day-old chickens, respectively, after primary vaccination (*Tran, 2016*). Differences in mean antibody titers reported in the present study and those reported in previous studies may be attributed to differences in schedules of vaccination and post-vaccination sample collection. Ducks in this study were vaccinated and sampled at an older age than those in other studies and, thus, produced stronger responses to primary vaccination.

Notably, booster immunization increased GMTs, reflecting the effect of vaccination. In addition, the proportion of ducks with high HI titers increased on every sampled farm. The results correspond to two studies conducted under field conditions in other countries. For instance, the previously mentioned study in Egypt revealed that the antibody titer levels markedly increased after the booster dose (*Kandeil et al., 2017*). Also, it was reported that H5-type virus vaccination increased seroconverted proportions after a booster vaccination in 13 member countries of the European Union (*Swayne, 2011*). These results are also consistent with findings of Bertelsen and Lecu et al. that suggested that the two-dose immunization regimen remarkably elevates the HI antibody titer levels in birds (*Bertelsen et al., 2007*; *Lecu et al., 2009*).

Because this study involved several duck farms, within- and between-farm variations in GMTs were observed (Fig. 1). First, ducks from the same farm showed different GMTs after primary vaccination because some ducks exhibited a seroconversion response, whereas some did not. This result may be attributed to various endogenous factors, such as differences in specific immune reaction, health status, or prevailing disease situation (*Marangon & Busani, 2007*; *Swayne, 2011*). Notably, this result may account for the broad range of vaccine-induced HI titers detected in this study, which corresponds with values reported by *Phan & Tran (2016)*. However, most ducks that failed to exhibit responses to primary vaccination showed seropositivity after booster vaccination. Veterinary authorities use the extent of variability of antibody response, which is commonly presented by %CV, as an index to evaluate the effectiveness of a vaccination program. For a majority of poultry diseases, %CV should not exceed 40% after a correct vaccine is administered (*Greenacre & Morishita, 2014*). High %CV values obtained in the present study provided evidence for considerable variation in antibody responses of ducks after primary vaccination. Some ducks showed high HI titers, whereas some showed low HI titers or even seronegativity. These results are consistent with findings of Tarigan et al. who reported that outcomes of field H5N1 vaccination were highly variable and farm-related. Specifically, HI titers of individual birds in each flock differed from those of birds in other flocks (*Tarigan et al., 2018*). Second, GMTs varied on the farm level (Fig. 1, Table 2); this

result may be attributed to differences in field conditions, which may be associated with environmental factors and rearing practices, immunization techniques, vaccine storage, vaccinator's skill and incentive, and other factors that vary across farms (*Tung et al., 2013*). Booster vaccination reduced within- and between-farm variation in antibody responses. Decreased variability in antibody responses following booster immunization has important implications in terms of the effectiveness of the vaccination program.

The most important goal of the H5N1 HPAI vaccination program is flock immunity, which is proportional to the level of protection achieved by all birds in a vaccinated flock. Achievement of flock-level immunity is used to evaluate the effectiveness of HPAI vaccination programs. In Vietnam, the national regulation stipulates that flock-level immunity is achieved if 70% of the poultry in each flock demonstrates seroconversion (HI titers $\geq 4$) and if 80% of the poultry flocks in each province or region shows flock-level immunity (MARD-DAH, Circular No. 07/2016/TT-BNNPTNT). In this study, primary and booster vaccinations provided some level of protection to most vaccinated ducks when the majority of antibody responses exceeded the cutoff level of $4\log_2$ (83.13% and 96.34% after primary and booster vaccinations, respectively). This finding may partly explain the fact that although local farmers often implement the single-dose vaccination regimen for their flocks, HPAI outbreaks have not occurred in the province since 2014 when the vaccine was first introduced. However, although seropositive rates considerably varied between farms, the overall seropositive rate achieved in this study at 21 days after primary vaccination (83%) was higher than that previously reported by *Pham (2015)* (68.18%) (*Pham, 2015*) and *Phan & Tran (2016)* (33.33%–40%) (*Phan & Tran, 2016*) for the same vaccine. *Henning et al. (2011)* and *Tung et al. (2013)* have reported low seropositive rates following primary vaccination with different strains.

Booster vaccination provided a higher level of immunity than primary vaccination. In all sampled farms in this study, booster vaccination produced higher seropositive rates than primary vaccination. The overall seropositive rate of more than 96% detected following booster immunization corresponds with the observation of provincial veterinary authorities in 2017 and reported rates by *Pham (2015)* and *Phan & Tran (2016)*. In terms of practical significance, results obtained after 21 days of booster vaccination may reflect the vaccine-induced serological immunity of ducks immediately before the common completion time of meat-type duck production cycle in the field, i.e., 63-day-old birds.

Although the minimum protective antibody titer of 1:16 ($4\log_2$) has been reported in Vietnam and four other countries worldwide (*Swayne, 2011*), the OIE Manual recommends that the minimum HI serological titer of birds under the field conditions should be 1:32 ($5\log_2$) for achieving a good probability of protection against mortality from HPAI infection (*OIE, 2015*). Nearly 85% of the vaccinated ducks in this study showed an antibody levels $\geq 5\log_2$ after booster vaccination, whereas approximately 68% showed these levels after primary vaccination. These results indicate that ducks undergoing booster vaccination have 17% increased probability of being protected from mortality in an outbreak. This finding was similar to previous reports that suggested that more than one vaccination dose is required to induce protective immunity and prevent H5N1 HPAI transmission in ducks and other poultry in field conditions (*Swayne, 2009*; *Van der Goot et al., 2007*;

*Pantin-Jackwood & Suarez, 2013*). In addition, *Lecu et al. (2009)* also demonstrated that the administration of booster vaccination to zoo birds in France is necessary to increase mean titers to a protective level.

Our findings are expected to reflect the current situation in the study area because we employed materials, serological assay procedures, expression and interpretation methods, and evaluation criteria similar to those used in the national post-vaccination surveillance program. Therefore, our results may guide veterinary authorities in Vietnam in their efforts to improve the effectiveness of the national H5N1 vaccination program.

## CONCLUSIONS

Primary and booster vaccinations are immunogenic and could induce antibody responses in ducks at levels that meet the targets of the national mass vaccination program. Our results support the notion that compared with the single-dose immunization regimen, the two-dose immunization regimen more intensely induces protective antibody production and, thus, provides better serological immunity against the HPAI virus in ducks. Furthermore, the single-dose vaccination regimen is suitable for short-lived meat ducks, whereas two-dose vaccination regimen is suitable for long-lived ducks, such as layers or breeders, to increase their protective humoral immunity and strengthen flock immunity. Further studies on the duration of antibody responses induced by the single-dose vaccination regimen are warranted. Furthermore, variations in antibody responses of vaccinated ducks suggest that the effectiveness of vaccination varies under different field conditions, which warrant additional attention.

## ACKNOWLEDGEMENTS

The authors gratefully acknowledge the Veterinary Diagnostic Laboratory and Treatment Division of the SDAH of Ben Tre Province, Vietnam for their collaboration in farm visits, sample collection, and laboratory work.

### Funding

This work was supported by a grant from Chiang Mai University. It was also supported by the Veterinary Public Health Centre for Asia Pacific (VPHCAP) and the Excellent Center of Veterinary Public Health, Faculty of Veterinary Medicine, Chiang Mai University. The funders had no role in study design, data collection and analysis, decision to publish, or preparation of the manuscript.

### Grant Disclosures

The following grant information was disclosed by the authors:
Chiang Mai University.
Veterinary Public Health Centre for Asia Pacific (VPHCAP).
Excellent Center of Veterinary Public Health.

Faculty of Veterinary Medicine.
Chiang Mai University.

## Competing Interests

The authors declare there are no competing interests.

## Author Contributions

- Hoa Thi Thanh Huynh conceived and designed the experiments, performed the experiments, analyzed the data, prepared figures and/or tables, authored or reviewed drafts of the paper, approved the final draft.
- Liem Tan Truong performed the experiments, contributed reagents/materials/analysis tools, farm contact and sampling.
- Tongkorn Meeyam contributed reagents/materials/analysis tools, additional revision of the manuscript.
- Hien Thanh Le conceived and designed the experiments, analyzed the data.
- Veerasak Punyapornwithaya conceived and designed the experiments, analyzed the data, prepared figures and/or tables, authored or reviewed drafts of the paper, approved the final draft.

## Animal Ethics

The following information was supplied relating to ethical approvals (i.e., approving body and any reference numbers):

This study was approved by the Institutional Ethical Review Board of Hanoi University of Public Health. The IRB was registered with the U.S. Department of Health and Human Services (IORG number 0003239, FWA number FWA00009326). IRB-HUPH approval number is 308/2017/YTCC-HD3.

## Data Availability

Huynh, Hoa Thi Thanh; Punyapornwithaya, Veerasak (2018): PeerJ #31719 ''Individual and flock immunity responses of naïve ducks on smallholder farms after vaccination with H5N1 Avian Influenza vaccine: a study in a province of the Mekong Delta, Vietnam''. figshare. Fileset. https://doi.org/10.6084/m9.figshare.7465202.v1.

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
