# Peer review of "Individual and flock immunity responses of naïve ducks on smallholder farms after vaccination with H5N1 Avian Influenza vaccine: a study in a province of the Mekong Delta, Vietnam"

_PeerJ, doi:10.7717/peerj.6268_

## Round 0.1 · original submission · Minor Revisions

In order to avoid data misinterpretation, it is important that you discuss the use of strain A\Ck\Scot\59 H5 as antigen in serological assay.

I think it is also important to include a discussion of other published studies that evaluated the effectiveness of the vaccine in inducing immunological responses under field conditions.

·

Basic reporting

no comment

Experimental design

no comment

Validity of the findings

no comment

Additional comments

The manuscript “#31719” by Huynh et al., entitled “Individual and flock immunity responses of naïve ducks on smallholder farms after vaccination with H5N1 Avian Influenza vaccine: a study in a province of the Mekong Delta, Vietnam” is a well-designed and well written account providing important information on humoral immune responses of ducks to the applied H5N1 vaccine under field conditions in Vietnam. This study is also essential for guiding the veterinary authorities, either locally in Vietnam or worldwide, to efficiently adopt effective booster vaccine protocol to vaccinate ducks which are commonly reared together with other avian species in hotspots areas of avian influenza viruses.
Generally, the English language of the manuscript is adequate; the quality of the figure and tables is satisfactory, the reference list cover the relevant literature adequately and in an objective manner. The topic is timely and interesting and the results are presented well.
However, the following points should be addressed:
1. Line 36-37 and line 107: remove “at 21-day intervals”. It is anyway stated in line 37 and may be confusing with the prevaccination sampling time point.
2. Line 57: replace “H5 HPAI viruses” with either “H5Nx HPAI viruses” or H5-type HPAI viruses” and keep it consistent throughout the manuscript (e.g. line 67, Line 102, line 123)
3. Line 64-65: the statement “The vaccine will continue to be used in the foreseeable future because it provides a certain protective effect against HPAI to the poultry population” is scientifically not optimum. Please phrase it for the present taking into consideration the rapid evolution of these H5-type strains from clade 2.3.2→2.3.2.1a→2.3.2.1b →2.3.2.1c and possible genetic and antigenic evolution into different subclades.
4. Line 73: keep distance between “H5N1” and “HPAI”.
5. Line 81-83: Concerning the statement “published data on the effectiveness of vaccines against avian influenza (AI), particularly the Re-6 vaccine, in inducing antibody response in domestic ducks under field conditions remain limited”, have the authors compared their data with similar studies in different geographical localities (e.g. Kandeil et al., 2017. Avian influenza H5N1 vaccination efficacy in Egyptian backyard poultry. Vaccine; 35 (45): 6195-6201). If not please discuss.
6. Line 93: replace “The Sample size” with Sample size.
7. Line 111: what other factors which can lead to H5-specific antibodies other than maternal antibodies or active infections of newly born duck. If no, please remove “or other unknown factors” and rephrase the statement accordingly.
8. Line 113: replace “which contains the HA gene of” with “which expresses the HA antigen of”.
9. Line 181: please provide a rationale before direct stating of the results (e.g. To investigate that maternal antibodies have already depleted, prevaccination HI titers for studied ducks were evaluated”.
10. Line 183: remove “with RAHO VI”.
11. Line 191: replace “produced” with “showed”.
12. Line 193: replace “H5N1 vaccine” with “Re-6 vaccine”.
13. Line 218-219: Have the authors checked or can check the effectiveness of induced antibodies titers to protect duck after first and booster vaccination doses (Challenge infection with currently circulating H5N1 HPAIV (Clade 2.3.2.1c))?
14. Line 240 and Line 298: discuss other studies which are performed under field or backyard setting in different geographical localities dealing with duck and other avian species` vaccination and adopting booster vaccination protocol.

·

Basic reporting

The manuscript by Huynh et. al. entitled “Individual and flock immunity responses of naïve ducks on smallholder farms after vaccination with H5N1 Avian Influenza vaccine: a study in a province of the Mekong Delta, Vietnam” is a straightforward to determine immunity responses against influenza H5N1 in duck smallholder farms after vaccination. vaccination has played a crucial role in the strategy for the prevention and control of H5N1 highly pathogenic avian influenza (HPAI). This study is essential for guiding the veterinary authorities.

Experimental design

The authors use well accepted methods in evaluating antibody responses of immunologically naïve domestic ducks to H5N1 avian influenza vaccine currently used in the national mass vaccination program of Vietnam.

Validity of the findings

The authors use well accepted methods in evaluating antibody responses of immunologically naïve domestic ducks to H5N1 avian influenza vaccine currently used in the national mass vaccination program of Vietnam.
However, there appears to be some careless misinterpretation of results, which needs checking and revision by the authors. Several hemagglutinin gene clades were detected in Vietnam clade 1.1.2 was predominant in 2012 and 2013 but gradually disappeared and clades 2.3.2.1a, 2.3.2.1b, and 2.3.2.1c from 2012 and recently clade 2.3.4.4 were detected. Why the authors used strain A\Ck\Scot\59 H5 as antigen in serological assay? And what the antigenic similarity between this strain and other strains circulated in Vietnam?

Additional comments

Minor Comments:
1- line 58; “…the failure of other anti-HPAI measures.” It is unclear the auther must explain what other measurements to control of avian influenza HP
2- line 108 “…after the primary vaccination (21 dpv), and 21 days after the booster vaccination (21 dpbv) Can the authors use consistent abbreviation in all manuscript.
3- In table 1 please include description for superscript letter a and b
4- In discussion, were there any published studies in literature on the evaluated the effectiveness of the vaccine in inducing immunological responses under field conditions? These should be discussed and compared to the result presented here.

---

## Round 0.2 · accepted · Accept

I can confirm that the revision has addressed all the points raised by the reviewers.

#